# Food Insecurity in Central-Eastern Europe: Does Gender Matter?

Hanna Dudek [1] and Joanna Myszkowska-Ryciak [2,*]

1 Department of Econometrics and Statistics, Institute of Economics and Finance, Warsaw University of Life Sciences (WULS), 02-776 Warsaw, Poland; hanna_dudek@sggw.edu.pl
2 Department of Dietetics, Institute of Human Nutrition Sciences, Warsaw University of Life Sciences (WULS), 02-776 Warsaw, Poland
* Correspondence: joanna_myszkowska_ryciak@sggw.edu.pl; Tel.: +48-22-5937022

**Abstract:** Food insecurity (FI) remains a challenge not only in less-developed countries but also worldwide. The literature indicates higher rates of FI for women than men in some regions of the world. Thus, the main objective of this cross-sectional study was to assess differences in experiencing FI according to gender in Central-Eastern Europe—a region that has been little researched in terms of FI. The study analyzes individual-level data on FI from the Gallup World Poll (GWP) for the years 2018–2019, obtained under a license from the Food and Agriculture Organization (FAO). Dataset encompasses representative samples of individuals aged 15 and above for each studied country. Apart from bivariate analysis, the ordered logistic regression, the generalized ordered logistic regression and multinomial logistic regression models were used. It was found that women experienced mild FI more often than men. However, gender differences with respect to moderate or severe FI were not confirmed. Moreover, the significant associations of all severity levels of FI with education, employment status, social capital, social network, age, marital status, household composition and income were observed. The research findings can be used to inform policy and to design targeted assistance programs for those in need.

**Keywords:** Sustainable Development Goals; Europe; socioeconomic and demographic characteristics; ordered logit model; generalized logistic regression; multinomial logistic regression

## 1. Introduction

As defined by the Food and Agriculture Organization of the United Nations (FAO), food insecurity (FI) occurs when individuals do not have adequate physical, social or economic access to sufficient, safe and nutritious food satisfying their nutritional requirements and food preferences for an active and healthy life [1]. Food insecurity is a substantial problem worldwide [2,3]. Therefore, the United Nations (UN) among the Sustainable Development Goals (SDGs) pointed out the need to "end hunger, achieve food security and improve nutrition and promote sustainable agriculture" [4]. To monitor Target 2.1 of the UN 2030 Agenda for SDGs, the prevalence of moderate or severe FI in the population, based on the Food Insecurity Experience Scale (FIES), has been used as the SDGs Indicator 2.1.2 [5].

Another of the SDGs integral to all dimensions of inclusive and sustainable development is women's equality and empowerment [6]. This equality should apply to all aspects of life and functioning in society, including especially food security (FS). However, the prevalence of moderate or severe food insecurity worldwide is slightly higher in women compared to men. At a global level, women had about a 13 per cent higher chance of being moderate or severe food insecure than men, and almost 27 per cent higher chance of experiencing severely FI. Two-thirds of countries worldwide reported higher rates of food insecurity for women than men [7]. Even when women have the same level of income, education and live in similar areas as men, their access to food is more difficult. It is worth

noting that gender gaps in poverty are the widest in the age of 25–34, which is the period of biological reproduction and childcare responsibilities [8].

A large body of evidence indicates that when women experience poverty, this negatively affects human capital. Poverty is a strong risk factor for FI, almost half of those living in poverty are food insecure [9]. When FS is disrupted, the nutritional value of the diet is initially reduced simultaneously with an increase in the share of energy, mainly from saturated fats and sugar, causing undernourishment and promoting excessive weight gain [10–13]. Evidence indicates that maternal undernutrition is associated with intrauterine growth restriction of fetus, with lifelong consequences for the future child's physical and mental development [14,15]. While obesity in women, especially during pregnancy, contributes to the health risks of their children and this deepens the health inequities across generations [16–18].

Conversely, women's greater access to income and resources, better nutritional status and higher education result in better health and educational outcomes for their children [19–21]. In turn, greater investment in child welfare improves the productivity of the next generation of workers and has a positive effect on economic development [22]. In addition, research shows that women tend to invest as much as 10 times more in their family's well-being, including in children's health, nutrition and education [23–25]. Consequently, when women control the household budget, family members tend to have better nutrition status, and children's survival rates increase [26]. It is worth emphasizing that addressing the dietary needs of adolescent girls, as well as women during pregnancy and lactation has been set as the Target 2.2 of the UN 2030 Agenda for SDGs [5].

Despite women's greater vulnerability to poverty, a low share of social protection policy is gender sensitive. Compared to men, women are more often involved in unpaid care and housework, which in turn limits their access to social protection [27]. Moreover, they are more likely to be working in low-paid sectors that do not offer sufficient social protection measures. When households cannot access adequate amount of food, this bias is likely to be reinforced, with negative consequences for the nutritional status and health of girls and women [28,29].

The FI status of a household or an individual is primarily influenced by economic but also sociodemographic factors and others, e.g., gender, employment skills, time, housing status, health status, food/cooking skills or capabilities, health insurance status, social support, past economic hardship and food accessibility [30,31]. Literature shows that these factors may be different depending on the country and/or region. This research focuses on gender differences in FI, which in context of the SDGs of the UN 2030 is particularly important.

### 1.1. Food Insecurity Assessment

Achievement of the SDGs largely depends on monitoring and follow-up processes [32,33]. Several methods and indicators are used to estimate FS and monitor its changes worldwide. The Food Insecurity Experience Scale (FIES) is an experience-based metric of food insecurity severity that ensures global comparability [34]. The FIES includes eight questions examining self-reported food-related behaviors and experiences associated with increasing difficulties in gaining access to food due to resource constraints of the individual respondent or of the entire respondent's household (Table 1). It is the official instrument used by the FAO to generate estimates of the prevalence of FI in the context of the SDGs' Target 2.1 monitoring [35].

**Table 1.** Questions in the FIES.

| No. | During the Last 12 Months, Was There a Time When, Because of Lack of Money or Other Resources: | Short Reference |
|---|---|---|
| (Q1) | You were worried you would not have enough food to eat | WORRIED |
| (Q2) | You were unable to eat healthy and nutritious food | HEALTHY |
| (Q3) | You ate only a few kinds of foods | FEWFOODS |
| (Q4) | You had to skip a meal | SKIPPED |
| (Q5) | You ate less than you thought you should | ATELESS |
| (Q6) | You ran out of food | RANOUT |
| (Q7) | You were hungry but did not eat | HUNGRY |
| (Q8) | You went without eating for a whole day | WHLDAY |

Own elaboration based on FAO [36].

The FIES is based on a well-established concept of FI experience consisting of three domains: worry/anxiety, changes in food quality and changes in food quantity [35,37]. With the FIES scale the risk of FI might be identified in communities and individuals in comparable manner in different populations. Based on the number of "yes" answers to questions (the FIES score,) the severity of FI can be accessed, ranging the FS status (zero positive answers) to all symptoms of FI (8 positive answers). FI is typically classified into four categories [35,38,39]:

1.  Food secure—raw scores of 0;
2.  Mild FI—raw scores of 1–3;
3.  Moderate FI—raw scores of 4–6;
4.  Severe FI—raw scores of 7–8 (see Figure 1).

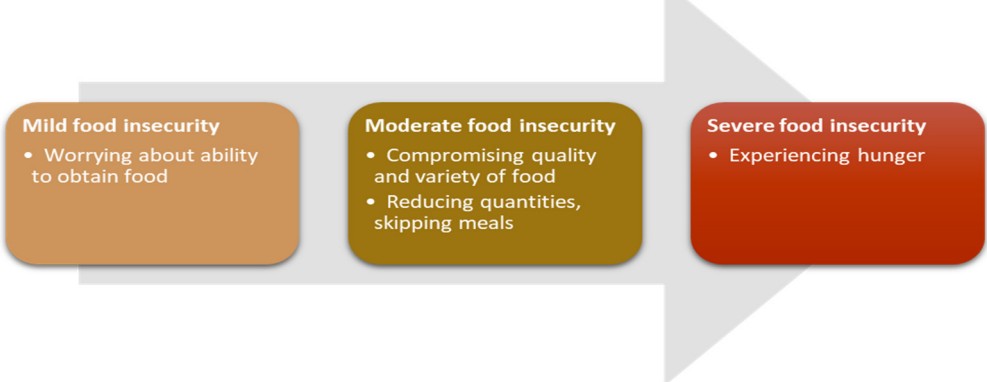

**Figure 1.** The severity range of food insecurity. Own elaboration based on FAO [36].

The FIES score analyzed in conjunction with the respondent and household characteristics can broaden the knowledge of FI risk factors and consequences on an individual and household level [40,41].

*1.2. Gender in Food Insecurity Research—A Brief Review of the Literature*

Most of the studies on FI include gender as one of the explanatory variables [40,42–44]. FI scores for women and men depend, among others, on: (i) country/region of residence and (ii) method of FI measurement.

Smith et al. [42] analyzed the FAO's FIES data from 134 countries from 2014 and showed different results for low-income, lower-middle-income, upper-middle-income and high-income economies. Broussard [45] investigating the FAO's FIES data from 2014 for 146 countries worldwide, presented the results for 11 groups of countries, which showed that significant differences in FI between women and men were not observed in all groups of countries. Similarly, Grimaccia and Naccarato [40], considering the FAO's FIES data for over 100 countries, obtained different conclusions depending on the analyzed group of countries. In particular, it was found that in intermediate, less-developed and in the least-developed countries, women experienced FI more often than men, while in very rich and developed countries, the opposite results were obtained [40].

Another issue is the method of measurement. Studies with the binary variable dominate the literature. Specifically, analyzing data based on responses to eight questions in the FIES about the individual's experience with food insecurity, the authors typically apply a cut-off of one out of eight [46,47], a cut-off of two out of eight [45], a cut-off of four out of eight [42,48] and a cut-off of seven out of eight [42,48].

The literature indicates that results regarding gender are sensitive to the chosen cut-off. For the threshold one out of eight [47], a higher prevalence of FI among Polish women than men has been observed. Similar results for the EU were revealed by Broussard [45] with the threshold two out of eight. However, no statistically significant differences at the 0.05 level

were found for moderate and severe FI in the EU. These results were also confirmed in the analysis of FI in 2017–2019 for Poland and Lithuania [49], where a higher mild FI among women than men was found, but no statistical difference referring to moderate or severe FI. Moreover, the choice of the model in the FI analysis is not without significance. The few studies using ordered logit models include Grimaccia and Naccarato [40] and Grimaccia and Naccarato [50]. They demonstrated that women experienced more FI compared to men—both globally and at the European level. In analyses where multinomial models were used, the results depend on whether mild, moderate or severe FI has been considered [49].

In addition to examining gender and FI, many studies also take into account various socioeconomic and demographic characteristics. Some factors influencing FI, such as poor education or low income, are universal in countries around the world [42]. Some of them, however, may be unique to a given country or a group of countries [40,42,49].

The few studies devoted to the relationship between FI and gender in the EU include Broussard [45] and Grimaccia and Naccarato [50]. Similar to Broussard [45] and Grimaccia and Naccarato [50], the study analyzed combined data from many countries belonging to a selected region. This analysis, however, concerned a rarely explored region: Central-Eastern Europe.

### 1.3. The Central-Eastern Europe Countries

The research focus was on eight post-communist countries of Central-Eastern Europe (CEE) who accessed the European Union (EU) in 2004. This EU enlargement incorporated the Visegrád Four (i.e., Czechia, Hungary, Poland and Slovakia) the Baltic Three (i.e., Estonia, Latvia and Lithuania) and Slovenia. The literature indicates, that despite the worse situation of these countries in 2004 compared to the "old" member states of the EU, the CEE countries have caught up the more advanced EU-15 economies [51,52].

The economies that acceded to the EU in 2004 have all had income levels below the EU average. However, analyzing the country-level FAO data it cannot be unequivocally said that the prevalence of moderate or severe FI in the "new" member states is higher than in the "old" member states of the EU (i.e., the countries that became the EU members before 2004) [53].

Since all these countries belong to the EU, food supplies as well as food quality and safety are subject to the same legal regulations and control systems. It is worth emphasizing that one of the EU's policy objectives is to ensure safe, nutritious, high-quality and affordable food for EU consumers [54]. However, as Hossain et al. [55] observed, food availability is not related to food accessibility, in other words, food supplies have minimal impacts on food security.

### 1.4. The Purpose and the Scope of the Study

The main goal of the study was to assess the differences in terms of different FI categories between women and men. Because of a very low incidence of severe FI, three categories: (i) food secure (FS), (ii) mild food insecure (MFI) and (iii) moderate or severe food insecure (SFI) were considered. To examine gender differences in food insecurity, bivariate and multivariate methods were applied. Apart from two-way findings (comparing FI and given individual factors), the results of the logistic regression models were presented. Such an approach allowed us to assess the significance of gender influence on FI after controlling typical socioeconomic and demographic characteristics indicated by the FI literature.

In this article statistical methods were used to address the following questions:

1. Are there any differences in experiencing FI according to gender in the CEE?
2. Are differences with respect to gender the same across all categories/severity levels of FI?
3. Are socioeconomic and demographic factors influencing FI with respect to gender the same across the CEE countries?

4. Do the results obtained with the use of different logistic regression models lead to the same conclusions?

Apart from bivariate methods, ordered, generalized ordered and multinomial logistic regression models were used. To the best of our knowledge, such "a bundle of models" has not been applied simultaneously so far in food insecurity analysis.

## 2. Materials and Methods

### 2.1. Description of the Dataset

This cross-sectional study uses individual-level data from the Gallup World Poll (GWP) (Global Research: See the World in Data | Gallup), made available by a license from the FAO. The GWP is a worldwide survey conducted annually in over 140 countries, using probability-based, multi-cluster sampling. It provides nationally representative samples of the adult population (aged 15 and above) in each country. More details about the GWP data in the context of FI research can be found in the FAO et al. report [7], Cafiero et al. [34], Ballard et al. [35], Broussard [45], Smith et al. [42] and dedicated website [36].

In this study, the dataset covering eight CEE countries (see Figure 2) was analyzed. Data relating to the situation before the COVID-19 pandemic have been taken into account. Due to the fact that for the Czech Republic and Slovenia the latest available data concerned year 2018, the data from 2018 and 2019 were combined. To be exact, the following data were included: for Estonia, Hungary, Latvia, Lithuania, Poland, Slovakia data from 2018–2019, for the Czech Republic and Slovenia—only from 2018.

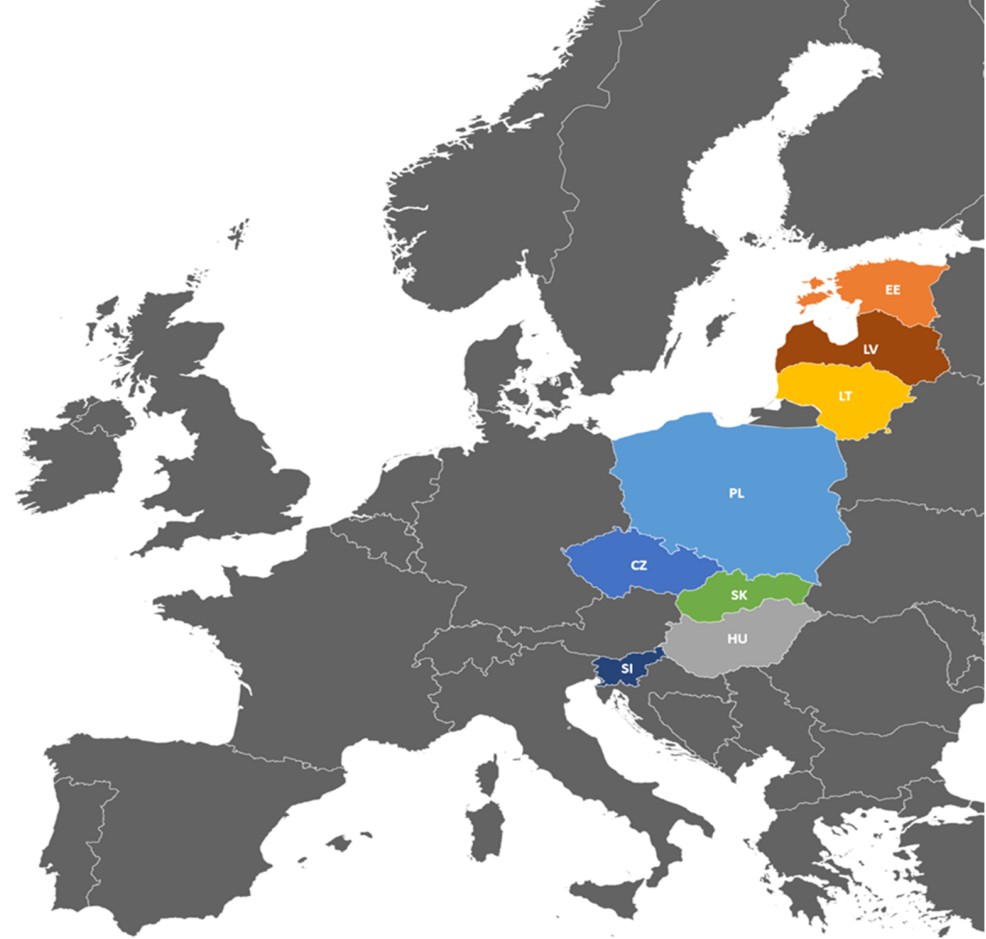

**Figure 2.** The location of analyzed CEE countries in Europe: Estonia (EE), Latvia (LT), Lithuania (LT), Poland (PL), Czech Republic (CZ), Slovakia (SK), Hungary (HU) and Slovenia (SI).

The sample size for a given year in a country was at least 1000 respondents aged 15 and above. In particular, in 2018 for the Czech Republic, Estonia, Hungary, Lithuania, Poland and Slovakia—1000 people; for Latvia—1021 people; and for Slovenia—2000 people. On the other hand, the 2019 data are samples of 1080 observations in each of the analyzed countries (i.e., Estonia, Hungary, Latvia, Lithuania, Poland, Slovakia).

Assessment of FI in this study is based on the Food Insecurity Experience Scale (FIES) that uses a set of eight questions capturing a range of FI, with yes/no response (see Table 1). Since in the analyzed CEE countries the prevalence of severe FI is negligible (approx. 1.5% of the population), in the study three categories were considered:

1.   Food secure with raw scores of 0 (FS);
2.   Mild FI with raw scores of 1–3 (MFI);
3.   Moderate or severe FI with raw scores of 4–8 (SFI).

Therefore, combined moderate and severe FI, which is of particular importance as it has been included in the SDG Indicator 2.1.2, was considered. Besides the FIES data, the Gallup World Poll database includes data relating to demographic and socioeconomic characteristics of individuals. Thus, the study examined the impact of various characteristics on FI. Apart from gender, the set of potential correlates included: educational level, age, the income quintile group and household composition. Moreover, the social capital and social network characteristics were considered. As in the work by Smith et al. [42], (i) social network is a binary variable that equals one if the respondent is satisfied with his/her ability to make friends, and (ii) social capital is a binary variable that equals one if the respondent feels she/he can count on friends and/or family in need. The household composition taking into account household size and presence in household at least 3 children up to 15 years old were included in the analyses. In the analysis of bivariate association, household size was categorized as a one-person, a two-person, a three-person, a four-person and at least a five-person household. Alternatively, in logistic regression models, logarithm of household size was also considered.

The educational level was categorized as elementary or lower (elementary), secondary and high or higher (tertiary). The age was classified into groups: below 34, 34–54, 55–69 and at least 70. Marital status was categorized as never married, married, living with partner, divorced or separated or widowed. The employment status was classified as fulltime employed for an employer, fulltime self-employed, out of workforce, part-time employee who wants to be fulltime employed and part-time employee who does not want fulltime employment.

*2.2. Methods*

As a first step, the proportions of the various FI categories for men and women were showed. Then, in this analysis, in order to analyze the strength of the relationship between respondents' characteristics and FIs among women and men, the Cramer's V coefficients were calculated. Finally, a regression model approach enabling the assessment of the significance of gender difference in respect to FI controlling socioeconomic and demographic factors was used.

In this study, logistic regression models to examine correlates of FI were used. As the outcome variable describing FI is ordered, the starting point in this research is ordinal logistic regression assuming (1):

$$P(y \leq j | \boldsymbol{x}) = \Lambda(\alpha_j - \boldsymbol{x}\boldsymbol{\beta}), \; j = 1, \, 2, \, \ldots, \, m, \tag{1}$$

where $\boldsymbol{x}$ is the vector of explanatory variables;
   $\alpha$ are the threshold parameters;
   $\boldsymbol{\beta}$ is the vector of the slope parameters;

$$\Lambda(\alpha_j - \boldsymbol{x}\boldsymbol{\beta}) = \frac{1}{1 + \exp(-(\alpha_j - \boldsymbol{x}\boldsymbol{\beta}))} \tag{2}$$

$m$ is the number of outcomes/categories.

The predicted probabilities belonging to a given category are defined as (3) [56,57]:

$$P(y_i = j|\boldsymbol{x}) = \Lambda(\alpha_j - \boldsymbol{x\beta}) - \Lambda(\alpha_{j-1} - \boldsymbol{x\beta}), \quad j = 1, 2, \ldots, m, \tag{3}$$

with $\alpha_0 = -\infty$ and $\alpha_m = \infty$.

In the study, three FI categories were considered, i.e., $m = 3$. Therefore, it is further proceeded as:

$$P(y = 1|\boldsymbol{x}) = \Lambda(\alpha_1 - \boldsymbol{x\beta})$$

$$P(y = 2|\boldsymbol{x}) = \Lambda(\alpha_2 - \boldsymbol{x\beta}) - \Lambda(\alpha_1 - \boldsymbol{x\beta})$$

$$P(y = 3|\boldsymbol{x}) = 1 - \Lambda(\alpha_2 - \boldsymbol{x\beta})$$

The ordinal logistic regression imposes a strong assumption of parallel regression. According to this assumption, the slope parameters should not differ for different categories. This assumption can be verified by the Brant test, the Wolfe–Gould test and the likelihood ratio test [56,58–61].

If the assumption of parallel regression is violated, such models as generalized ordered logit model or multinomial model can be used [57]. The generalized ordered logit model can be written as (4):

$$P(y \leq j|\boldsymbol{x}) = \Lambda(\alpha_j - \boldsymbol{x\beta}_j) \tag{4}$$

or:

$$P(y_i = j|\boldsymbol{x}) = \Lambda(\alpha_j - \boldsymbol{x\beta}_j) - \Lambda(\alpha_{j-1} - \boldsymbol{x\beta}_j), \quad j = 1, 2, \ldots, m, \tag{5}$$

with $\alpha_0 = -\infty$ and $\alpha_m = \infty$ and $\Lambda$ described by Formula (2).

Thus, the generalized ordered logit model allows the slope parameters to differ for each category $j = 1, 2, \ldots, m$ [56,57]. Therefore, Formula (4) generalizes (1) and Formula (5) generalizes (3).

A multinomial logit model is defined for nominal outcome. However, it is often used for ordinal data [56,62]. The multinomial logit model can be expressed as (6):

$$P(y = j|\boldsymbol{x}) = \frac{\exp\left(\alpha_j + \boldsymbol{x\beta}_{j|b}\right)}{\sum_{r=1}^{m} \exp(\alpha_r + \boldsymbol{x\beta}_{r|b})} \tag{6}$$

where $b$ is the base (reference) category.

In this study, the outcome of one is used as the base category. Therefore, the predicted probabilities are calculated as (7):

$$P(y = j|\boldsymbol{x}) = \begin{cases} \frac{1}{1 + \sum_{r=2}^{3} \exp(\alpha_r + \boldsymbol{x\beta}_r)} & \text{for } j = 1 \\ \frac{\exp\left(\alpha_j + \boldsymbol{x\beta}_j\right)}{1 + \sum_{r=2}^{3} \exp(\alpha_r + \boldsymbol{x\beta}_r)} & \text{for } j = 2 \text{ or } 3 \end{cases} \tag{7}$$

The parameters of all described models were estimated by maximizing the log-likelihood. Statistical analyzes were performed using the STATA program (StataCorp LP, College Station, TX, USA).

In this research, the analyzed outcome variable has three categories: 1 denotes food security (FS), 2 means mild food insecurity (MFI) and 3 denotes moderate or severe food insecurity (SFI). For deeper investigation of the implication of the violation of parallel regression assumption, all described models were applied.

## 3. Results

The analytical sample comprised 15,501 individuals (52.88% female) spanning o eight CEE countries for the period of 2018–2019. The following sections present the prevalence of FI status in terms of gender, the associations between FI and the characteristics of respondents and the results of the logistic regression models.

### 3.1. The Prevalence of Food Insecurity

Table 2 presents the differences of the prevalence of FS status in terms of gender in CEE countries.

**Table 2.** Gender differences in FS status in CEE countries.

| Gender | Percent | Std. Error | 95% Confidence Interval | |
|---|---|---|---|---|
| | | Food secure | | |
| Men | 81.45 | 0.54 | 80.36 | 82.49 |
| Women | 78.07 | 0.51 | 77.06 | 79.05 |
| | | Mild food insecure | | |
| Men | 13.13 | 0.48 | 12.23 | 14.09 |
| Women | 15.31 | 0.44 | 14.47 | 16.19 |
| | | Moderate or severe food insecure | | |
| Men | 5.42 | 0.32 | 4.83 | 6.07 |
| Women | 6.62 | 0.31 | 6.03 | 7.26 |

Source: Own elaboration.

When analyzing the results presented in Table 2, it can be noticed that men experienced food security more often than women (corresponding 95% confidence intervals do not overlap). However, when considering FI, only significant differences were found with regard to mild FI.

A comparison between women and men in terms of answers to individual FIES questions is presented in Figure 3. These eight questions (see Table 2) focus on the respondents' behaviors and experiences related to the increasing difficulty in accessing food as a result of resource constraints.

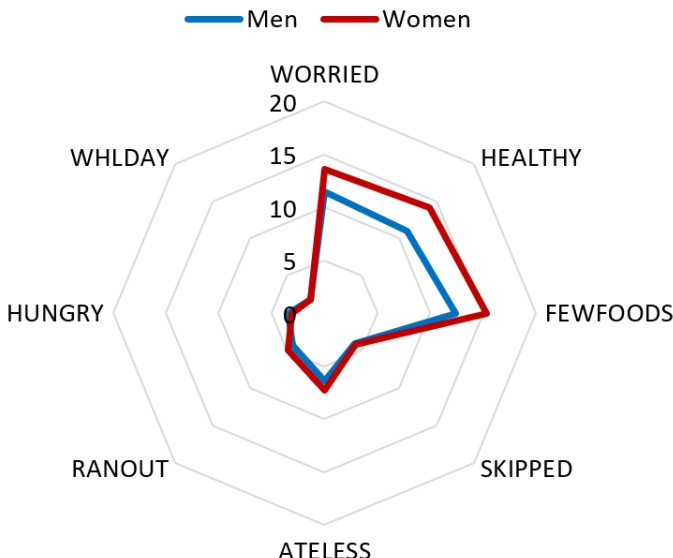

**Figure 3.** Percentage of positive responses to individual FIES questions (Table 2).

Figure 3 provides a detailed insight into the differences in responses to individual FIES questions.

On the basis of Figure 3, it can be seen that women more often than men answered positively to the questions Q1–Q3. These questions relate to mild FI [63]. However, the differences in the answers to the questions Q4–Q8 are not so pronounced. This means that in terms of experiencing moderate or severe FI, the response rate of women does not differ much from that of men.

### 3.2. Food Insecurity and Respondent Characteristics—Bivariate Analyses

To assess the association between FI and respondent characteristics, bivariate analyses were performed (Table 3). In this case, three categories of FI were also included.

**Table 3.** Association of food insecurity with respondent characteristics.

| Variable | $\chi^2$ Statistics | Cramer's V |
|---|---|---|
| Gender | **34.602** | **0.047** |
| Age | **57.708** | **0.043** |
| Household size | **214.411** | **0.083** |
| At least three children | **95.926** | **0.079** |
| Marital status | **377.988** | **0.111** |
| Education | **344.206** | **0.106** |
| Employment status | **366.382** | **0.109** |
| Social network | **166.206** | **0.073** |
| Social capital | **232.379** | **0.123** |
| Income quintile | **1300** | **0.202** |

Note: numbers marked in bold indicate results that are significant at a 0.05 level.

For all variables, significant bivariate associations between FI and socioeconomic and demographic characteristics have been found. These results indicate that income and social capital were the factors with the closest relationship with FI, while the characteristics with a weaker relationship with FI were gender and age. Thus, a relationship between FI and gender was weak, but significant.

To assess the gender differences regarding strength of the dependence, Cramer's V coefficients were calculated separately for men and women (Figure 4 and Table 4).

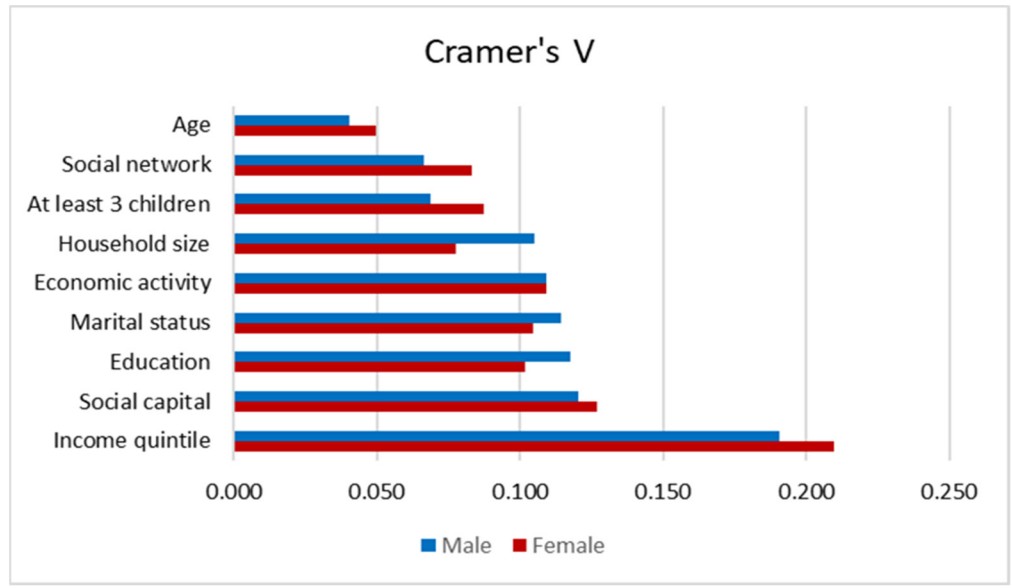

**Figure 4.** Association between food insecurity and the respondents' characteristics (Cramer's V measures).

On the basis of Figure 4, it can be concluded that there are minor gender differences between Cramer's V values. For example, considering the social network, its relationship with FI is greater among women. Contrarily, for household size, a higher Cramer's V was recorded for men; wherein household size was categorized as a one-person, a two-person, a three-person, a four-person and at least a five-person household. The significance of these differences was verified by determining the confidence intervals with bootstrapping. This enabled us to assess whether the nature of the dependence of the individual's characteristics

and FI was the same for women and men. Detailed information on the differences in the Cramer's V measure among women and men is presented in Table 4.

**Table 4.** Cramer's V values and their 95% confidence intervals.

| Gender | Cramer's V | Std. Error | [95% Confidence Interval] | |
|---|---|---|---|---|
| | | Age | | |
| Women | 0.050 | 0.007 | 0.032 | 0.061 |
| Men | 0.040 | 0.010 | 0.020 | 0.051 |
| | | Social network | | |
| Women | 0.083 | 0.008 | 0.066 | 0.095 |
| Men | 0.066 | 0.010 | 0.047 | 0.077 |
| | | At least three children | | |
| Women | 0.087 | 0.015 | 0.060 | 0.121 |
| Men | 0.068 | 0.016 | 0.037 | 0.099 |
| | | Household size | | |
| Women | 0.077 | 0.008 | 0.059 | 0.089 |
| Men | 0.105 | 0.010 | 0.081 | 0.123 |
| | Economic activity (employment status) | | | |
| Women | 0.109 | 0.009 | 0.085 | 0.123 |
| Men | 0.109 | 0.010 | 0.086 | 0.126 |
| | | Marital status | | |
| Women | 0.105 | 0.007 | 0.089 | 0.116 |
| Men | 0.114 | 0.009 | 0.093 | 0.128 |
| | | Education | | |
| Women | 0.102 | 0.007 | 0.087 | 0.115 |
| Men | 0.118 | 0.010 | 0.096 | 0.138 |
| | | Social capital | | |
| Women | 0.127 | 0.013 | 0.101 | 0.149 |
| Men | 0.120 | 0.015 | 0.091 | 0.147 |
| | | Income groups | | |
| Women | 0.210 | 0.008 | 0.195 | 0.224 |
| Men | 0.191 | 0.009 | 0.171 | 0.207 |

Source: Own elaboration.

Based on the results presented in Table 4, as for each characteristic corresponding 95% confidence intervals for women and men overlap, it can be stated that there were no significant differences in the strength of FI dependency and individual characteristics among women and men. Therefore, the inclusion of all these characteristics in the logistic regression models seems to be justified.

*3.3. Logistic Regression Models Results*

In the next stage of the study, the logistic regression model was used to assess differences across gender in the EU controlling socioeconomic and demographic factors influencing FI. Firstly, the assumption of parallel regression for the ordered logit model was verified. Results shown in Table 5 indicate strong rejection of this assumption.

**Table 5.** The results of tests of parallel regression assumption.

| Test | Chi-Square Statistics | df | $\chi^2$ (0.05; 30) | *p*-Value |
|---|---|---|---|---|
| Brant | 133.1 | 30 | 43.773 | 0.000 |
| Wolfe–Gould | 129.2 | 30 | 43.773 | 0.000 |
| Likelihood ratio | 125.7 | 30 | 43.773 | 0.000 |

The results in Table 5 therefore indicate that the ordered logit model is not appropriate for these data. In such a situation, generalized ordered or/and multinomial logit models should be considered. However, in order to compare the results for all three models given by Formulas (3), (5) and (6), their estimates are presented in Table 6. For the ordered logit

model, one set of coefficients for all FI severity levels was estimated; however, this was different for the multinomial and generalized logit models. In the multinomial logit model, the first set of coefficients refers to mild FI (MFI) versus food secure (FS), the second set of coefficients compares moderate or severe food insecure (SFI) to food secure (FS). In the generalized logit model, the first set of estimates refers to any level of FI versus FS, the second set of estimates compares moderate or severe food insecure (SFI) to mild FI (MFI) or food secure (FS). Table 6 presents the basic results of the models' estimation. Results that are significant at a 0.05 level are marked in bold. Detailed results of the model estimation are presented in the Supplementary Materials.

**Table 6.** Ordered, multinomial and generalized ordered logit models' results. Country fixed effects are included in models.

| Variable | Ordered | Multinomial | | Generalized Ordered | |
|---|---|---|---|---|---|
| | | MFI vs. FS | SFI vs. FS | FI vs. FS | SFI vs. MFI or FS |
| Women | **0.136** | **0.126** | 0.170 | **0.135** | 0.166 |
| Log HHsize | **−0.674** | **−0.505** | **−1.009** | **−0.640** | **−0.871** |
| At least 3 children | **0.665** | **0.451** | **0.964** | **0.622** | **0.851** |
| Social capital | **−0.645** | **−0.331** | **−1.056** | **−0.553** | **−0.914** |
| Social network | **−0.205** | **−0.254** | −0.117 | **−0.216** | −0.104 |
| Education (ref. elementary) | | | | | |
| Tertiary | **−0.872** | **−0.660** | **−1.371** | **−0.831** | **−1.256** |
| Secondary | **−0.420** | **−0.302** | **−0.582** | **−0.393** | **−0.495** |
| Age (ref. at least 70) | | | | | |
| Below 34 | **0.320** | 0.200 | **0.537** | **0.293** | **0.474** |
| Age 34−54 | **0.447** | **0.240** | **0.789** | **0.404** | **0.706** |
| Age 55−69 | **0.329** | **0.277** | **0.406** | **0.322** | **0.327** |
| Marital status (ref. single (never been married)) | | | | | |
| Married | **0.001** | **0.001** | 0.0005 | **0.001** | 0.0003 |
| Widowed | **0.443** | **0.473** | **0.458** | **0.464** | **0.335** |
| Divorced or separated | **0.597** | **0.453** | **0.839** | **0.571** | **0.701** |
| Domestic partner | **0.324** | **0.288** | **0.435** | **0.313** | **0.380** |
| Employment status (ref. unemployed) | | | | | |
| Employed fulltime for an employer | **−0.553** | **−0.498** | **−0.725** | **−0.610** | **−0.541** |
| Fulltime self-employed | **−0.527** | **−0.762** | −0.285 | **−0.689** | −0.110 |
| Out of workforce | **−0.619** | **−0.614** | **−0.711** | **−0.687** | **−0.542** |
| Part-time employee (want) | 0.052 | 0.067 | 0.008 | 0.024 | −0.091 |
| Part-time employee (do not want) | **−0.556** | **−0.796** | −0.292 | **−0.691** | −0.099 |
| Income quintile group (ref. first quintile group) | | | | | |
| Second quint. group | **−0.786** | **−0.548** | **−1.160** | **−0.738** | **−0.975** |
| Third quint. group | **−1.269** | **−0.999** | **−1.786** | **−1.224** | **−1.490** |
| Fourth quint. group | **−1.537** | **−1.299** | **−1.966** | **−1.496** | **−1.659** |
| Fifth quint. group | **−2.118** | **−1.766** | **−2.876** | **−2.063** | **−2.518** |
| Cut1 | **−0.683** | - | - | - | - |
| Cut2 | **0.868** | - | - | - | - |
| Constant | - | −0.018 | −0.272 | **0.641** | −0.891 |

When analyzing the results concerning gender significance, it can be pointed that different results for the ordered logit model and for the other two models were obtained. Ordered logit models results indicated statistically significant difference between women and men in all FI categories (severity levels). This was not confirmed by the results of multinomial and generalized ordered logit models, according to which moderate or severe food insecurity among women does not significantly differ than among men. However, it should be underlined that the results presented in Table 5 provide evidence that the parallel regression assumption was violated. Thus, the use of ordered logit model leads to the misleading conclusion regarding the gender differences.

The same is true for the social network—no statistically significant difference has been found here in terms of moderate or severe food insecurity. On the other hand, the significance of social capital and education was recorded in all FI categories. When it comes to household composition, household size and at least three children turned out to have a significant impact on FI.

Taking into account age, a higher probability of FI among people under the age of 70 than among people over 70 was generally found. Marital status turned out to be an important differentiating factor for FI. Compared to single (never been married), other individuals experienced FI more often. The only exception were married respondents, where there was no significant difference in moderate or severe food insecurity.

The status in the labor market turned out to be an influential factor. In particular, fulltime employed for an employer and those being out of workforce were less FI than unemployed. However, no significant difference between fulltime self-employed and unemployed regarding moderate or severe food insecurity was found. Moreover, as expected, belonging to a specific income group was important for IF. Individuals from the higher quintile groups were less vulnerable to FI than those being in the lowest quintile group.

## 4. Discussion

Food insecurity in CEE is caused not only by low income, but also by other overlapping issues. As some studies indicate, FI also applies to people who are not poor [43]. Thus, there is a need to identify various socioeconomic and demographic correlates of FI for a better understanding of the problem.

### 4.1. Gender and Food Insecurity

The main focus in this study is gender differences regarding experiencing FI in various categories (severity levels). To explore this, first bivariate analyses were performed. Applying chi-square test, it was found that gender is a factor that significantly differentiates FI. In addition, investigating the prevalence of food security and mild food insecurity, a better situation for men compared to women was demonstrated. However, analyzing moderate or severe FI, no difference with respect to gender was found. It should be emphasized, however, that the differences in CEE are smaller than in some regions of the world, e.g., Latin America [7,36,45].

During the next stage, gender differences in food insecurity controlled for socioeconomic and demographic characteristics were examined. To investigate this issue, the ordered logit model, generalized ordered logit model and multinomial logit model were used. The application of multinomial and generalized ordered models lead to similar conclusions. On the other hand, the results for gender differences found via the ordered model differ from those obtained using the other models. However, it has been shown that the ordered logit model does not meet the assumption of parallel regression. Therefore, inference about the role of individual characteristics should be made on the basis of multinomial and/or generalized ordered models. It is worth mentioning that many authors do not verify this assumption. This can lead to misleading conclusions regarding the association of FI with socioeconomic and demographic factors. Specifically, the results of ordered logit model estimation indicate that women are more moderate or severe food insecure than men. However, this was not confirmed by the results obtained from the multinomial and generalized ordered models.

One of the reasons why women may experience greater mild FI compared to men is that women are primarily responsible for day-to-day food supply decisions and food provisioning in their households [64,65]. Broussard [45] suggests that they may therefore be more aware of the problems in meeting their food needs before these problems become serious. Gender inequalities, still deep-rooted in many societies, might be a contributing factor [6,26]. Despite the fact that women are increasingly involved in paid work, the share of domestic tasks and care work is still not equal. This results in a double burden for women and, consequently, greater susceptibility to stressful situations [66]. Additionally,

women tend to experience higher levels of anxiety, frustration and depression than male when reacting to stress [67]. As Broussard suggests [45], men because of shame, pride or other reasons might not be so willing to report less severe experiences compared to women.

The findings regarding the relationship between FI and gender are consistent with those of Broussard [45] revealing the gender difference with respect to mild FI in the EU but there was no significant difference at the 0.05 level regarding moderate and severe FI. However, the study results differ from those obtained by Grimaccia and Naccarato [40,50] who applied ordered logit. In the above-mentioned studies, women experienced more FI compared to men. Perhaps this discrepancy is the result of the ordinal logistic regression model used by the authors [40,50].

The obtained results are important for the SDGs for gender equality. In the context of sustainable development, ensuring food security, including the proper nutritional status of girls and women, is of key importance [6]. Food security disruptions, especially in middle- and upper-income countries, are usually compensated by choosing foods with a lower nutritional value, which leads to excessive body weight gain and qualitative malnutrition [68]. FI might be associated with poorer diet quality and health status [69]. Anemia, caused by poor nutrition and deficiencies of iron and other micronutrients, contributes to maternal mortality and low birth weight [70]. Food insecurity can increase the likelihood of pregnancy complications and have a direct impact on the fetus development and the future health of both woman and child [71]. This can increase healthcare costs, hinder active participation in the labor market and consequently and have a negative effect on economic development [22].

### 4.2. Other Correlates of Food Insecurity

The other results of the present study are largely in line with empirical studies that use FAO's FIES data [42,45,50]. In accordance with previously mentioned authors [42,45,50], it was found that experiencing FI was associated with low levels of education. This may be due to the fact that better educated people are more aware of the importance of lifestyle, especially diet, for overall health and well-being. Consistently with Smith et al. [42], the study findings confirmed the positive impact of social capital in reducing the risk of experiencing FI. Thus, feeling that people could count on help from friends and/or family in need was an informal insurance. When it comes to social network, Broussard [45] and Smith et al. [42] found its dependence on all severity levels of FI. However, its significance in terms of experiencing moderate or severe FI has not been confirmed. The results only indicated that being satisfied with the ability to make friends matters in the mild FI experience. Furthermore, marital status was a characteristic playing an important role in the likelihood of experiencing FI. As Grimaccia and Naccarato [50], this study found that separated or divorced respondents more often experienced FI (on all FI severity levels) compared to single individuals. Contrary to Grimaccia and Naccarato's [50] results for the whole Europe, our findings indicated that single individuals were not more affected by mild FI than married respondents. However, as Grimaccia and Naccarato found, the region of Europe may matter in this regard. Specifically, they showed that the difference between these two groups in Eastern Europe was not significant at the 0.05 level. In addition, widowed and domestic partners were taken into account and the worse situation of these individuals compared to single individuals was presented.

The findings of Smith et al. [42] confirmed that the role of employment status depends on the FI severity level. Moreover, in line with Broussard [45], it was found that fulltime employed and out of the workforce were less vulnerable to FI than the unemployed. However, in the present study, employed fulltime for an employer and fulltime self-employed were additionally distinct. Furthermore, unlike Broussard [45], a part-time employee who wants be fulltime employed and a part-time employee who does not want fulltime employment were considered separately and their different FI situation compared to the unemployed was shown.

Age was another characteristic considered in the present work. A significant but weak relationship between age and FI was showed. It is difficult to directly compare our results with those of other studies in this regard. In the analyzed works, a slightly different way of including age in the models was adopted. Especially, Grimaccia and Naccarato [40,50] and Smith et al. [42] showed an inverted U-shape relationship between the age of the respondent and FI. This means that the most vulnerable to FI were middle-aged people. This result was largely confirmed in the present study, as a higher FI for people aged 34–59 compared to people 70 was found.

It is worth underlining the importance of our results on FI in the context of the COVID-19 pandemic. Recent research in countries such as Brazil [72], Mexico [38], the United States [73] and Poland [74] revealed that COVID-19 and national lockdowns have a substantial impact on FI status. Additionally, there is some evidence that women remain disproportionately affected by the socioeconomic fallout during the COVID-19 pandemic, struggling with the loss of jobs and livelihoods, disrupted education and increased burdens of unpaid care work. Between 2019 and 2020, women, who were already underrepresented in employment and the labor force, suffered steeper job losses than men. As women earn less, save less and are the majority of single-parent households, their capacity to respond to the economic crisis is therefore less than that of men [75]. Unfortunately, data from 2020 and 2021 are not available to us. Nevertheless, in order to be able to understand the impact of the pandemic on FI, it is essential to have a reliable baseline for comparison. Thus, in the light of the COVID-19 pandemic, the present results contribute by providing a baseline for comparing the experience of pre- and post-pandemic FS in CEE in future research. However, we are aware that the issue of FI requires continuous monitoring, which is a premise for further exploration in this area.

### 4.3. Study Strengths and Limitations

The research aimed to explore differences in the prevalence of FI between men and women in CEE, and to investigate the role of socioeconomic and demographic factors among women and men. The study adds to the understanding of the gender association with FI status (severity levels) in Central-Eastern Europe. It provides a cross-sectional analysis of a survey conducted by GWP. It explores this issue using FIES data validated worldwide by the FAO. These data include nationally representative samples of the population 15 years of age and older. Such data give the opportunity to identify the characteristics of population groups at a greater risk of FI. According to the FAO's recommendations, the severity of food insecurity was classified as mild, moderate or severe FI [36]. Due to a very low prevalence of severe FI, it was combined with a moderate FI, and finally three categories, food secure, mild FI, moderate or severe FI, were analyzed. It enabled us to distinguish between serious and less serious conditions of food insecurity. The differences between the categories are important for research as well as policy purposes.

It was showed that the results in terms of gender are sensitive to the modelling approach. Therefore, it is also important to check the assumptions used in the models. Apart from bivariate analysis, the study applies ordered, generalized ordered and multinomial logit models. In the present study, the assumption of parallel regressions imposed on ordered logit models has been rejected. Therefore, a multinomial logit model and a generalized ordered logit model were used. The conclusions from these two models are very similar—thus, the findings are robust.

This study is not without limitations. In FI analysis, it would be interesting to take into account the impact of other characteristics apart from those included in this research. For example, the individual situation with regard to the burdening of household income with fixed expenses (e.g., loan repayments, fixed house maintenance fees) might be crucial. It would also be worth taking into consideration the biological type of the household in which the respondent lives (i.e., whether the household consists of parents and children, or if it is a multi-generation unit, etc.). Unfortunately, the GWP do not provide detailed information on the above-mentioned issues.

Finally, it should be point out that these results are for combined data from several countries. It should be stressed that they may differ slightly from one country to another. However, the goal of the present study was to show the importance and complexity of FI with respect to gender in the wider CEE perspective.

## 5. Conclusions

The study provides a new insight into the gender–food insecurity relationship in Central-Eastern Europe. A slightly higher rate of mild FI reported by women compared to men was found. However, the present study highlights that the differences between women and men in terms of FI are not as large as in some regions of the world, e.g., Latin America, as no significant gender difference in moderate or severe FI was observed. In addition, after stratification by gender, it was demonstrated that economic activity, income, education, social network, social capital, household composition, marital status and age were significantly associated with women's as well as men's FI status. Thus, the mentioned factors influenced both women and men FI. It was observed that FI prevalence decreased with increasing severity. Importantly, the present study demonstrated that the results in terms of gender are sensitive to the modelling approach.

The identification of groups particularly vulnerable to FI may allow for the appropriate targeting of countermeasures against FI and their adaptation to people with different demographic, social and economic characteristics. Deeper understanding of the factors that are associated with FI should help to target the most vulnerable FI individuals as well as orientate policy recommendations.

The study results indicate that the policy should be particularly targeted at households with at least three children, individuals with low education, the unemployed and the lowest income groups. The present study highlights the important role of social capital. Hence, a large role to play for various social welfare centers serving not only material assistance but also psychological support, to ensure people that they have someone to count on when in need. Collaboration between government agencies and non-government organizations, including food banks, can be beneficial to reduce FI. Efforts to improve food security in CEE should be undertaken through policy. It would be useful to monitor the situation at different FI severity levels and not be limited only to people experiencing severe FI. Early intervention among mild FI individuals, with particular emphasis on women, could prevent the problem from worsening.

Policy should, first and foremost, promote healthy and sustainable consumption behaviors, support vulnerable households and individuals and reduce income poverty. However, to reinforce these findings there is a strong need for increased scope of research using more recent data and a wider range of explanatory variables. Further research should focus on the changes caused by the pandemic and the Russia–Ukraine war. Especially, an important aspect is the analysis of FI among people particularly hit by the effects of the COVID-19 pandemic. A new issue to be explored is the prevalence of FI among Ukrainian refugees, the largest group of which are women.

**Supplementary Materials:** The following supporting information can be downloaded at: https://www.mdpi.com/article/10.3390/su14095435/s1, Table S1. The ordered logit model results; Table S2. The multinomial logit model results; Table S3. The generalized ordered logit model results.

**Author Contributions:** Conceptualization, H.D. and J.M.-R.; methodology, H.D.; formal analysis, H.D.; data curation, H.D.; writing—original draft preparation, H.D. and J.M.-R.; writing—review and editing, J.M.-R. All authors have read and agreed to the published version of the manuscript.

**Funding:** This research received no external funding.

**Institutional Review Board Statement:** Not applicable.

**Informed Consent Statement:** Not applicable.

**Acknowledgments:** The GWP data access was granted by the FAO through a competitive call for proposals. The authors are grateful to Meghan Miller from the FAO Food Security and Nutrition Statistics Team for her assistance and kind support.

**Conflicts of Interest:** The authors declare no conflict of interest.

## Glossary

| | |
|---|---|
| CEE | Central and Eastern Europe |
| FAO | Food and Agriculture Organization |
| FI | Food insecurity |
| FIES | Food Insecurity Experience Scale |
| FS | Food security |
| GWP | Gallup World Poll |
| MFI | Mild food insecurity |
| SDGs | Sustainable Development Goals |
| SFI | Moderate or severe food insecurity |

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
