# Peer review of "Food Insecurity in Central-Eastern Europe: Does Gender Matter?"

_sustainability, doi:10.3390/su14095435_

Round 1

Reviewer 1 Report

Title of the manuscript: Food insecurity in Central-Eastern Europe: does gender matter?

Manuscript ID: sustainability-1652596

This study assessed the differences in terms of food insecurity categories between women and men. Moreover, this study utilized the FAO's Food Insecurity Experience Scale, which was set as an official metric for assessing one of the SDGs related to reduction of hunger. The findings of this study showed women experienced mild food insecurity more often than men. Overall, this study addresses a topic of high relevance for research and also for practice. However, I believe some issues need revision and clarification. Addressing the below comments help improving this manuscript:      

General comments

  1. The English grammar and style should be checked throughout the manuscript.
  2. The authors should avoid using pronouns such as “we”, “our” and “us” in the text.
  3. Given that the study presents a long list of abbreviations, I suggest adding a “glossary” table at the end of the paper as it will aid the readers to learn about the concepts/terms that they are about to study.

Abstract

  1. In the Abstract section, the objectives of the study are not mentioned clearly; this point is very important and should be outlined clearly.
  2. A few policy implications based on results should be added at the end of the abstract in 1-2 sentences.
  3. The author should avoid repeating keywords that already exist in the title (e.g., food insecurity; gender; Central-Eastern Europe). I suggest replacing them with other words that have been mentioned in the text; also, 6 keywords are enough.

Introduction

  1. Lines 51-52 in page 2: “Poverty is a strong risk factor for FI: almost half of those living in poverty are food insecure [11].Ë® The authors should remove colon ":" in the sentence.
  2. Throughout the manuscript, in some sentences the authors use Sustainable Development Goals and sometimes its abbreviation (SDGs). Please be consistency in the whole text.
  3. Lines 128-130 in page 4: “Smith et al. [46], analyzing the FAO’s FIES data from countries from 2014, showed different results for low-income, lower middle-income, upper-middle-income and high-income economies.Ë® The authors should modify “analyzingË® to “analyzedË®.
  4. Overall, in the Introduction section, the organization of the inputs is good; however, this section needs to first identify the “gap” in the literature, present a compelling argument that why the “gap” needs to be filled, and convince readers that their approach is appropriate and practical. In short, the study needs to convince readers that how exactly this study contributes to the relevant literature.

Methodology

  1. Line 200 in page 5: In the subsection “2.1. The DataË® the authors should modify it to “Data collectionË® or “Data analysisË®.
  2. The authors should cite the number of equations in the main text.
  3. Lines 279-280 in page 7: “A multinomial logit model is defined for nominal outcome. However, it is often used for ordinal data [Error! Reference source not found.,64].Ë® The authors should modify the underline part of the sentence.
  4. In the method section, equations should be define properly.

Results

  1. In the Results section, I believe it would help readers better follow the presentation of results if there was a short introduction paragraph to state how the section is organized.
  2. Tables and Figures in the Results section needs to be discussed in more details based on the findings of the study.

Discussion

  1. In this section, inputs need to be supported with sufficient and relevant references. The findings and their implications should be discussed in the broadest context possible and limitations of the work. Moreover, the authors should compare/contrast their findings with similar studies (2017-2022).

Conclusion

  1. The Conclusion section needs to be enriched significantly. In this section, authors should directly discuss the main implications of findings and avoid presenting those concluding remarks that were already mentioned in the previous sections. More importantly, there should be some highlights on how this study is going to be beneficial to the policy makers in 1-2 paragraphs.
  2. The authors should highlight the specific and practical suggestions with respect to their findings at the end of the Conclusion section in one paragraph.
  3. The future research direction should be added to the end of the Conclusion section.

Reviewer 2 Report

Food insecurity  is a challenge  in less-developed countries and worldwide, especially women, which has the significant association of its all severity levels with education, employment status, social capital, social network, age, marital status, household composition and income. These results are of great theoretical significance and practical value for improving food security. The analysis process is comprehensive, good organized, large amount of information and so on. Minor revision can be published in Sustainability. However, there are some major issues need to be improved:

  1. Keywords:Method of the keywords can be reduced;
  2. Introduction: Food safety in the Central-Eastern Europe should be highly outlined and supplemented in the preface;
  3. Results: Figure or Table tries to be arranged on the same page;Can the reasons analyze the research results.
  4. Discussions: Increasing discussion on the contribution of high-resistant starch rice and barley and its grass powder to human health and food health safety is beneficial to expand readership, it is recommended for reference
  • https://www.tandfonline.com/doi/full/10.1080/87559129.2021.2024221

https://www.hindawi.com/journals/omcl/2020/3836172/ 

Reviewer 3 Report

General comments

It is an exciting study that proposes to analyze the food insecurity (FI) situation in Central-Eastern Europe. Important issue aligned to the United Nations Sustainable Development Goals (SDGs) achievement. However, to present that, the authors need to reconsider the paper. Below is shown some aspects to help the revision of the material.

Article type needs to be informed.

Abstract:  Besides the mentioned information, the authors should present details about the setting, participants, study design, data used and variables considered in the analytical procedure, not overcoming the 200-word limit proposed by the journal instruction.

Key-words: Should be selected following the journal recommendation (number and its meaning considering the study proposal)

Introduction

Line 5: it seems that something is missing

Line 84: It is missing a link between lines 84 -85

The introduction needs to be reduced and edited to present an overview of the paper's main aspects.

Headings and sections should follow the research article type as stated by the journal author's instruction. In addition, the STROBE guide could also help.

Methods:

Methods did not clearly show details about the data source and permission to use (e.g., secondary data derived from; selected by; organized) and precise definition of outcomes, exposures, predictors, potential confounders, and effect modifiers as well as variables of interest, sources of data and details of methods of assessment. 

Results and discussion

Results and discussion need revision according to the recommendations presented in the previous sections.

Suggested reference:

von Elm E, Altman DG, Egger M, Pocock SJ, Gotzsche PC, Vandenbroucke JP. The Strengthening the Reporting of Observational Studies in Epidemiology (STROBE) Statement: guidelines for reporting observational studies. Ann Intern Med. 2007; 147(8):573-577. PMID: 17938396

Round 2

Reviewer 1 Report

The authors have addressed my comments carefully and my major remarks from the previous version are addressed sufficiently. In my opinion, the content and structure of this study can be suitable for publication in the journal of Sustainability. However, there are a few minor points need to be addressed.

Reviewer 3 Report

The results presented by this study are of great significance and give interesting insights into improving food security. The revisions in the text of the manuscript improved its quality allowing its recommendation for publication.

Author Response

Thank you very much for your time and constructive and helpful comments.